| 1  | Contribution of dark inorganic carbon fixation to bacterial carbon demand in                      |
|----|---------------------------------------------------------------------------------------------------|
| 2  | the oligotrophic Southeastern Mediterranean Sea                                                   |
| 3  | Tom Reich 1,2*, Natalia Belkin 1, Guy Sisma-Ventura 1, Hagar Hauzer 1, Maxim                      |
| 4  | Rubin-Blum <sup>1,2</sup> , Ilana Berman-Frank <sup>2*</sup> , and Eyal Rahav <sup>1,3,4*</sup>   |
| 5  |                                                                                                   |
| 6  | <sup>1</sup> National Institute of Oceanography, Israel Oceanographic and Limnological            |
| 7  | Research, Haifa, Israel.                                                                          |
| 8  | <sup>2</sup> Department of Marine Biology, Leon H. Charney School of Marine Sciences,             |
| 9  | University of Haifa, Haifa, Israel.                                                               |
| 10 | <sup>3</sup> Department of Earth and Environmental Science, Ben-Gurion University of the          |
| 11 | Negev, Beer Sheva, Israel.                                                                        |
| 12 | <sup>4</sup> Institute of Marine Science, University of California, Santa Cruz, CA, USA.          |
| 13 |                                                                                                   |
| 14 | *Correspondence: TR- <u>treich02@campus.haifa.ac.il</u> ; IBF- <u>iberman2@univ.haifa.ac.il</u> ; |
| 15 | ER- eyal.rahav@ocean.org.il                                                                       |
| 16 |                                                                                                   |
| 17 | Abstract                                                                                          |
| 18 | Photosynthetically derived organic matter sinking to depth from the illuminated layers            |
| 19 | is often not sufficient to meet the energy demands of microbes in the dark ocean. This            |
| 20 | 'mismatch' is especially notable in the warm and oligotrophic eastern Mediterranean               |
| 21 | Sea where the annual photosynthesis is one of the lowest in the world's oceans, yet its           |
| 22 | aphotic zone is considered a hotspot for microbial activity and biomass. Here, we                 |
| 23 | investigated the role of photic and aphotic dark inorganic carbon fixation rates (DCF)            |

and its contribution to bacterial carbon demand in the southeastern Mediterranean Sea during the mixed and stratified periods. Our results demonstrate that DCF rates are measurable throughout the water column (0-1750 m) and are on the same order of magnitude as photosynthesis (34 vs. 45 g C m<sup>-2</sup> y<sup>-1</sup>, respectively). Using a carbon mass balance that considers photosynthesis, DCF and bacterial production (and hence respiration) we show that chemoautotrophy provides ~35% of the 'missing carbon' supply needed for microbial growth and activity in the aphotic layer, while other sources of dissolved organic carbon remain to be elucidated. These findings underscore the need for further research into the factors affecting DCF, its role in global carbon budgets, and its potential to enhance atmospheric inorganic carbon sequestration.

### 1 Introduction

The oceans aphotic layers contain the world's largest reservoir of dissolved inorganic carbon (DIC) (Baltar et al., 2010; Burd et al., 2010; Reinthaler et al., 2010), and harbor ~65% of all prokaryotes (Whitman et al., 1998). Aphotic prokaryotes typically rely on utilization of organic matter (and carbon), fixed by photoautotrophs via photosynthesis and exported from the euphotic zone, to sustain their growth and accumulate biomass (del Giorgio and Duarte, 2002). Current estimates reveal, however, a discrepancy between the supply of particulate organic carbon from photosynthesis and the bacterial organic carbon demand (BCD) in the aphotic zones (Ducklow, 2000; Karl et al., 1988; Smith and Azam, 1992). This mismatch suggests that there are other source/s of carbon that are being utilized by aphotic microorganisms (Baltar et al., 2009; Herndl and Reinthaler, 2013). One such source, that has been far less investigated, involves the fixation of DIC by chemo-autotrophic microbes and its assimilation into new biomass

(Baltar and Herndl, 2019). This could subsequently provide bioavailable DOC to other 49 microbial populations at depth (Baltar et al., 2010).

DIC uptake by heterotrophic bacterioplankton is generally attributed to anaplerotic reactions (Dijkhuizen and Harder, 1984; Erb, 2011) which are metabolic pathways that replenish intermediates enzymes in the citric acid cycle by fixing CO<sub>2</sub>, but other microorganisms such as nitrifying bacteria can also fix DIC (Alonso-Sáez et al., 2010). Genomic studies on deep-sea microbial communities identified several genes and metabolic pathways that enable some microbes to thrive as chemoautotrophs on inorganic substrates (Berg et al., 2007; Hallam et al., 2006). Measurements of CO<sub>2</sub> fixation by chemoautotrophs and heterotrophic bacterioplankton are scarce, yet substantial dark DIC fixation (DCF) rates have been reported in various oceanic settings and water masses (Swan et al. 2011; Zhou et al. 2017; La Cono et al. 2018; Alothman et al. 2023) and maybe more common than previously thought (Hansman et al., 2009; Herndl et al., 2005). The deep waters of the southeast Mediterranean Sea are characterized by higher concentrations of inorganic nutrients compared to the photic zone (e.g., ~6 µmol NO<sub>3</sub>+NO<sub>2</sub> kg<sup>-1</sup> and ~0.2 PO<sub>4</sub> μmol kg<sup>-1</sup> (Ben-Ezra et al., 2021; Sisma-Ventura et al., 2021) and low bioavailable dissolved organic carbon (Martínez-Pérez et al., 2017; Santinelli, 2015; Santinelli et al., 2010). Despite these characteristics, the southeast Mediterranean Sea's aphotic waters are considered a hotspot for bacterial activity compared to other oceanic regimes at similar depths (Luna et al., 2012; Rahav et al., 2019). Nutrient addition bioassays and water mixing simulations suggest that aphotic prokaryotes are primarily carbon-limited (Hazan et al., 2018; Rahav et al., 2019). Here, we report on both photic and aphotic DCF and heterotrophic bacterial production

rates from 6 cruises held between 2021-2023 in the southeastern Mediterranean (bottom

depth 1500-1750 m) during the mixed (winter) and stratified (summer) periods. Our results demonstrate that DCF rates cannot be neglected (contrary to past convention, Nielsen 1952) and are within the same order of magnitude as photosynthesis or heterotrophic bacterial production (BP). We also show that DCF substantially contributes to bacterial carbon demand (BCD), therefore providing, some of the 'missing carbon' supply needed for microbial growth and activity in the aphotic layer of the southeast Mediterranean Sea.

### 2 Material and methods

2.2 Sample collection - Seawater was collected during six seasonal cruises in the Levantine Basin, southeast Mediterranean Sea, on-board the R/V Bat-Galim between 2021-2023. Three cruises were held during the stratified period and three during the winter mixing. The mixed layer depth was calculated using a temperature difference of Δ0.3 °C (Mena et al., 2019). Two 'deep' stations were sampled in each cruise; one located at the edge of the continental shelf (H05 33.00 Lat, 34.50 Lon, bottom depth ~1500 m, 50 Km from the coast) and the other at the edge of Israel's exclusive economic zone (H06 33.15 Lat, 34.16 Lon, bottom depth 1750 m, 90 Km from shoreline). Seawater was sampled at discrete depths throughout the water column, from the surface (~0.5 m) to the bottom (1500-1750 m) using Niskin bottles. Sampling depths were chosen in real-time based on reads of Conductivity Temperature Depth (CTD) (Seabird 19 Plus), chlorophyll fluorescence (Turner designs, Cyclops-7) and PAR (Sea Bird). The raw hydrological data can be freely downloaded from https://isramar.ocean.org.il/isramar2009/. Measurements included DIC (NaH14CO<sub>3</sub>) uptake under ambient light (hereafter 'light primary productivity, LPP) or under full dark conditions (DCF), bacterial productivity (BP) and nutrient quantification.

Nitrite and ammonium concentrations – Samples for nitrite (NO<sub>2</sub>-) and ammonium 98 (NH<sub>4</sub><sup>+</sup>) concentrations were collected only in the 2023 cruises. The samples were pre-99 100 filtered (0.45 μm), placed in acid-washed plastic vials, and were kept frozen at -20 °C until analysis. Nutrients were measured with a Seal Analytical AA-3 system. The limits 101 of detection for NO<sub>2</sub><sup>-</sup> and NH<sub>4</sub><sup>+</sup> were 0.06 μM and 0.09 μM, respectively. 102 103 2.3 LPP and DCF - Seawater was collected in triplicates into transparent (for LPP measurements) or dark (for DCF) Nalgene bottles (45-250 ml) and spiked with 104 NaH<sup>14</sup>CO<sub>3</sub> (Perkin Elmer, specific activity 56 mCi mmol<sup>-1</sup>) following Nielsen, (1952). 105 The bottles were maintained in on-deck incubators covered with a gradient of neutral 106 mesh simulating the irradiance intensity (no change in spectrum) at 100%, 50%, 10%, 107 1%, and 0.1% of surface light intensities or under complete dark conditions (Belkin et 108 al., 2022; Reich et al., 2024). Incubators were kept at constant ambient surface 109 temperatures (~19-20 °C in winter and ~28-29 °C in the summer cruises). We 110 111 acknowledge that temperature differences between surface and deeper depths may alter 112 the LPP or DCF rates measurements, especially during the summer when the water column is stratified. While in situ measurements may offer more precise rate estimates, 113 they are generally impractical during research cruises that involve sampling at multiple 114 locations and times throughout the day and night. Nevertheless, preliminary 115 comparisons between the incubation setup used here versus in situ incubations using a 116 mooring line showed negligible differences in primary productivity, falling within the 117 expected range of measurement variability (see also Reich et al., 2022). All the 118 119 incubation bottles were spiked at sunrise and terminated after 24 h (Reich et al., 2022; 120 Robinson et al., 2009) by filtering, the particulate matter onto GF/F filters using low vacuum pressure (<50 mmHg). Next, the excess <sup>14</sup>C-bicarbonate was removed by 121 fuming with 50 µl of 37% hydrochloric acid overnight. Finally, 5 ml scintillation 122

cocktail (ULTIMA-GOLD) was added, and the disintegrations per minute (DPM) from the particulate matter concentrated on the filters were counted using a TRI-CARB 4810 TR (Packard) liquid scintillation counter. Blank seawater spiked with NaH<sup>14</sup>CO<sub>3</sub> was filtered immediately without incubation and the reads were subtracted from the sample's DPM. The blank DPM reads were usually negligible (<5% of the sample's DPM). Aliquots (50 µl) from random spiked samples were placed onto new GFF filters, added with 50 µl ethanolamine and scintillation liquid, and counted immediately without incubation to account for the 'added activity' of the radiolabeled working solution used. LPP was calculated as the difference between the DPM retrieved from the samples incubated under ambient light ('total primary production) and the 'dark' bottles. Dark or light dissolved inorganic carbon fixation was calculated based on the Bermuda Atlantic Time-series Study (BATS) protocol (https://bios.asu.edu/bats/batsdata). More details can be found in Reich et al. (2024). 2.3 Bacterial production- Triplicate samples per depth (1.7 ml) were incubated in the dark with 10 nmol/L <sup>3</sup>H-leucine L<sup>-1</sup> (Perkin Elmer, specific activity 123 Ci mmol<sup>-1</sup>) for 4-6 h under ambient temperature (Simon et al., 1990). The incubations were terminated with 100 µl of trichloroacetic acid (100%), processed as described by Smith and Azam (1992), and counted using a TRI-CARB 4810 TR (Packard) liquid scintillation counter. Killed control samples containing <sup>3</sup>H-leucine L<sup>-1</sup> and trichloroacetic acid (without incubation) were also measured and these control sample's DPMs were subtracted from the sample's reads. A conversion factor of 3 kg C mol<sup>-1</sup> per mole leucine incorporated was used, assuming an isotopic dilution of 2.0 (Simon and Azam 1989). 2.4 Molecular analyses and statistics - DNA was extracted from water samples with the PowerWater kit (Qiagen, USA), using the FastPrep-24<sup>TM</sup> Classic (MP Biomedicals,

USA) bead-beating to disrupt the cells (2 cycles at 5.5 m sec-1, with a 5 min interval). 147 The V4 region (~ 300 bp) of the 16S rRNA gene was amplified from the DNA (~50 148 149 ng) using the 515Fc/806Rc primers amended with relevant tags (Apprill et al., 2015; Parada et al., 2016). PCR conditions were as follows: initial denaturation at 94 °C for 150 45 s, 30 cycles of denaturation (94 °C for 15 sec), annealing (15 cycles at 50 °C and 15 151 cycles at 60 °C for 20 sec) and extension (72 °C for 30 s). Two annealing temperatures 152 153 were used to account for the melting temperature of both forward (58.5-65.5 °C), and reverse (46.9-54.5 °C), primers. 154 Demultiplexed paired-end reads were processed in QIIME2 V2022.2 environment 155 (Bolyen et al., 2019). Reads were truncated based on quality plots, checked for 156 chimeras, merged and grouped into amplicon sequence variants (ASVs) with DADA2 157 (Callahan et al., 2016), as implemented in QIIME2. The amplicons were classified with 158 Scikit-Learn classifier that was trained on Silva database v138 (16S rRNA, (Glöckner 159 160 et al., 2017). Mitochondrial and chloroplast sequences were removed from the 16S rRNA amplicon dataset. Downstream analyses were performed in R v4.1.1 (R Core 161 Team, 2021), using packages Phyloseq (McMurdie and Holmes, 2013) and Ampvis2 162 (Andersen et al., 2018). Indicator species analyses were performed using Indic species 163 package v1.7.9 (De Ca'ceres et al., 2009). Amplicon reads were deposited to the NCBI 164 SRA archive under project number PRJNA1215023. 165 166 2.5 Bacterial respiration (BR), bacterial carbon demand (BCD) and zooplankton 167 respiration (ZR) - BR was calculated based on the following equation and assuming an 168 average open-ocean bacterial growth efficiency (BGE) of 20% (Herndl and Reinthaler, 169 2013) similar to previous direct measurements from the Mediterranean Sea ranging 170 171 from 0.21-0.29 (Zweifel et al., 1993).

$$BGE = \frac{BP}{BP + BR}$$

BCD was then calculated as the sum of BP and BP (Gasol et al., 1998). Zooplankton respiration (ZR) and excretion were compiled from Belkin et al. (2022).

### 3 Results and discussion

3.1 Dark and light inorganic carbon fixation rates – As expected, LPP was restricted to the photic layer with highest rates usually measured at the surface (~0.5 m) that gradually decreased to reach minimum rates at the bottom of the photic layer (~180 m) (Fig 1A). Relatively low LPP values were measured during the stratified summer (~0.1-0.8 µg C L<sup>-1</sup> d<sup>-1</sup>), whereas higher rates were measured during the winter mixing period (~0.2-7.4 μg C L<sup>-1</sup> d<sup>-1</sup>) (Fig 1A). This resulted in ~10-fold higher integrated rates measured during the mixed period compared to those measured during the stratified period (Table 1), in accordance with studies from the area (Psarra et al., 2005; Reich et al., 2022; Sisma-Ventura et al., 2022b). In contrast with LPP, DCF was not restricted to the photic layer and ranged from 0 to ~0.4 µg C L<sup>-1</sup> d<sup>-1</sup> throughout the water column (Fig 1B, Fig 2A), without significant differences in the absolute rates between the photic and aphotic zones (t-test, p>0.05, Fig 2B). The observed decrease in DCF rates with depth (Figure 2A) during the summer cruises may be partly attributed to a decline in the abundance of chemoautotrophs with depth. For example, Agogué et al., (2008) reported a decline in archaeal amoA gene copy numbers with depth in the eastern North Atlantic. Normalizing DCF rates to chemoautotrophic microbial cell abundance (or gene copy) could reveal a different vertical pattern. Another possible explanation for the decline in DCF rates with depth may be related to the weakening flux of sinking organic matter with depth that limits the substrates that fuel DCF (discussion below). The integrated photic DCF was typically lower than the rates reported in the central and

western Mediterranean Sea (La Cono et al., 2018). The aphotic DCF rates were ~3.5 fold higher during the mixed than during the stratified period (Table 1, Fig 2A).

**Figure 1**: Spatial and temporal variability in rates of LPP (a), DCF (b), BP (c) and the contribution of DCF to bacterial carbon demand (BCD) (d) at the offshore SE Mediterranean Sea (Lat. 33.15 N, Lon. 34.16 E) between 2021-2023. BCD was calculated assuming a bacterial gross efficiency of 0.20 (Gasol et al., 1998).

**Table 1**: Integrated rates and contribution of DCF to metabolic processes in the photic (0-180) and aphotic (>180 m) depths of the pelagic southeast Mediterranean Sea. The values represent the minimum and maximum ranges observed across the cruises, with the averages and their corresponding standard deviations provided in parentheses. BDL = Below detection limit.

| Variable                                    | Season     | Photic (0-180 m)          | Aphotic<br>(180-1750 m)  |
|---------------------------------------------|------------|---------------------------|--------------------------|
|                                             | Mixed      | $158-649$ $(368 \pm 205)$ | BDL                      |
| LPP (mg C $m^{-2} d^{-1}$ )                 | Stratified | 4-69                      | BDL                      |
|                                             |            | $(32\pm 26)$              |                          |
| DCF (mg C m <sup>-2</sup> d <sup>-1</sup> ) | Mixed      | $6-27$ $(15\pm 8)$        | $17-342$ $(152 \pm 127)$ |

|                                            | Stratified | 7-19<br>(14±5)          | 8-127 (59 ± 48)         |
|--------------------------------------------|------------|-------------------------|-------------------------|
| DD (ma C m-2 d-1)                          | Mixed      | $6-58$ $(28\pm21)$      | $12-65$ $(33\pm 22)$    |
| BP (mg C m <sup>-2</sup> d <sup>-1</sup> ) | Stratified | $9-55$ $(30\pm20)$      | 7-123 $(81 \pm 55)$     |
| DCE 4.11 (* 4. DCD (0/) *                  | Mixed      | $23-221$ $(109 \pm 88)$ | $49-594  (213 \pm 200)$ |
| DCF contribution to BCD (%) *              | Stratified | 8-31<br>(18±9)          | 8-42 (23 ± 13)          |
| DCE 11 (1 1 1 1 1 1 DD (0/)                | Mixed      | 1-15<br>(6±5)           |                         |
| DCF contribution to total PP (%)           | Stratified | $12-81  (40 \pm 32)$    |                         |

<sup>\*</sup> Assuming bacterial gross efficiency of 0.2 (Gasol et al., 1998) and that the available DOC for bacteria is 20% of the total primary productivity at the photic layer (Teira et al., 2003).

The higher aphotic DCF in the mixed versus the stratified periods may be related to more bioavailable carbon that is transported from the photic layer as marine snow and supplies organic carbon to heterotrophic activity in the winter (coinciding higher LPP). However, given the oligotrophic nature of the southeast Mediterranean Sea (Berman-Frank and Rahav, 2012; Reich et al., 2022), most of the organic carbon (both particulate and dissolved originating from LPP) is recycled within the photic layer. Only a small fraction fluxes down to the aphotic zone and has been recorded in sediment traps (Alkalay et al., 2024).

3.2 Bacterial productivity in relation to DCF and BCD - Another possible mechanism that may, potentially, explain higher DCF rates during the winter versus the summer is anaplerosis. The extent of anaplerotic reactions is primarily driven by the availability of labile organic carbon to heterotrophs (Dijkhuizen and Harder 1984). Therefore, assuming anaplerosis drives DCF, we expect it will be positively coupled to BP.

Yet, our results do not support the likelihood of significant anaplerosis reactions, predominantly evident from the spatiotemporal distribution of aphotic BP (Fig 1C) differing considerably from that of the DCF (Fig 1B) and does not correlate with it (Fig 3A). In fact, BP seems to be coupled with LPP at the photic layer reaching ~0.4  $\mu$ g C L<sup>-1</sup> d<sup>-1</sup> (not shown). Excluding some sporadic measurements, aphotic BP rates were usually of similar magnitude and typically <0.1  $\mu$ g C L<sup>-1</sup> d<sup>-1</sup> (Fig 1C). Moreover, the highest integrated aphotic BP was measured during the summers of 2021 and 2022 and not during the winter cruises when generally higher DCF was recorded (Table 1, Fig 2A).

**Figure 2**: Dark carbon fixation rates in the southeastern Mediterranean Sea. Averaged vertical distribution of DCF in the offshore southeast Mediterranean Sea during the mixed (white) and stratified (gray) periods (a), and a box plot showing the DCF rates at the photic (0-180 m) and aphotic (>180 m) water depths (b).

Despite the lack of a clear positive relationship between DCF and BP, DCF may contribute to bacterial carbon demand (BCD) in the aphotic zone. Thus, we use a

literature standard, a bacterial growth efficiency of 0.20 (Gasol et al., 1998) to calculate BR and BCD (see the 'material and methods' section for more details). This calculation yielded bacterial respiration (BR) ranging from 29-494 mg C m<sup>-2</sup> d<sup>-1</sup> (average 209±172 mg C m<sup>-2</sup> d<sup>-1</sup>), and the concurrent BCD ranges from 36 to 648 mg C m<sup>-2</sup> d<sup>-1</sup> (average 262±121 mg C m<sup>-2</sup> d<sup>-1</sup>). Under these circumstances, exudation of DOC from primary productivity at the photic layer estimated as 20% of the rates (Teira et al., 2003) equals to  $\sim 1-130 \text{ mg C m}^{-2} \text{ d}^{-1}$ . This new DOC, that originated form the photic zone therefore cannot support the aphotic BCD in our system in all of our observations. However, if we consider the contribution of DOC produced by aphotic DCF, part of the missing carbon may be accounted for. Thus, when considering aphotic DCF in addition to the sequestered DOC from the photic layer, the 'abnormally high' aphotic BCD could be explained in full  $(\ge 100\%)$  in  $\sim 35\%$  of the observations (Fig 1D). In the other 65% of the observations the missing carbon sources needed to support the aphotic BCD remains an enigma. We note that these calculations are based on global averages and assumptions and therefore may be subject to some uncertainties. For example, BGE can vary between seasons and sites (del Giorgio and Cole, 1998). In the Mediterranean Sea, long-term measurements of BGE ranged from 0.21 (similar to our calculations and the global average used by Herndl and Reinthaler, 2013) to 0.29 (Zweifel et al., 1993). If the 0.29 value is used, the contribution of DCF to the aphotic BCD increases to ~45% of the observations rather than ~35% when using BGE of 0.2. Similarly, if we apply an exported DOC estimate of ~4% from the photic zone, as reported for the Ionian Sea/western Mediterranean (Moutin and Raimbault, 2002), the relative contribution of DCF to aphotic BCD would be even higher than in our current calculations, which assume ~20% DOC export (Teira et al., 2003). These uncertainties warrant future investigation.

Yet, even when using conservative estimates for BGE and DOC export as done here, 269 the contribution of DCF to aphotic BCD remains substantial. 270 271 Evidence suggests that dissolved methane may be more abundant in oxygenated 272 environments than previously thought (Grossart et al., 2011). Methane can potentially be one of the missing energy sources for marine microbes and support high BCD 273 274 (Brankovits et al., 2017) as observed at the aphotic southeast Mediterranean Sea (Fig. 275 1D). In agreement, methanotrophs were found in aphotic cold seeps at the southeast Mediterranean Sea (Sisma-Ventura et al., 2022a), as well as across the aphotic water 276 277 column in our samples (see discussion below). 3.3 Interannual variability in aphotic DCF - Interannual variability in DCF, but not in 278 LPP or BP, was observed with higher rates recorded in March 2021-March 2022 and 279 280 lower rates observed in August 2022-August 2023 (Fig 1B). Inorganic nutrients such as PO<sub>4</sub><sup>3+</sup> or NO<sub>3</sub><sup>-</sup>+NO<sub>2</sub><sup>-</sup> are unlikely to explain this variability as their ambient levels 281 similar 282 were between periods (https://isramar.ocean.org.il/isramar2009/). Alternatively, we surmise that differences in the bioavailability and concentration of 283 sinking organic particles, possibly attributed by the BiOS (Bimodal Oscillating System) 284 oscillation circulation of deep water between the Adriatic and Ionian seas, could 285 potentially explain the higher aphotic chemoautotrophic activity in March 2021-March 286 2022 versus August 2022-August 2023. This mechanism is known to influence the 287 bioavailability of organic nutrients in the deep Mediterranean Sea by modulating deep-288 water circulation and ventilation patterns (Civitarese et al., 2010). These shifts affect 289 the transport and residence time of organic matter (Civitarese et al., 2023), thereby 290 291 potentially altering availability of organic nutrients to aphotic microbial populations, including to chemoautotrophs. Supporting this hypothesis are recent studies from the 292 northern Red Sea and South China Sea showing that DCF is limited by labile organic 293

nutrients such as phosphonates and even carbon-rich molecules (Reich et al., 2024; Zhou et al., 2017). Aphotic free-living chemoautotrophs are likely to encounter an increasingly refractory pool of dissolved organic matter for metabolism that may result in lower DCF rates, as shown in exported material through the water column (Santinelli et al., 2013). Particle-attached chemoautotrophs may have access to higher concentrations of organic substrates. Therefore we surmise these microbes would preferentially have a particle-attached lifestyle in the deep ocean. The patchy nature of particulate matter sinking and lateral transport during wintertime (Alkalay et al., 2024) and aggregate concentrations (Bar-Zeev et al., 2012) in the deep southeast Mediterranean Sea could also potentially explain the interannual variability in DCF between periods. Understanding how chemoautotrophs transform labile dissolved organic matter into refractory dissolved organic matter, which is an essential process in the 'microbial carbon pump', is crucial as it influences the efficiency of the biological pump (Jiao et al., 2010). Oxidizing reduced inorganic compounds as electron donors (e.g., NO<sub>2</sub>- or NH<sub>4</sub>+) may provide chemoautotrophic prokaryotes sufficient energy to fix DIC (Hügler and Sievert, 2011). We therefore measured the vertical distribution of NO<sub>2</sub>- and NH<sub>4</sub>+ (only in the March and August 2023 cruises) and examined if these chemical species are coupled or uncoupled with DCF at the aphotic zone. Our results show a negative-linear relationship between DCF and NO<sub>2</sub>- (Fig 3B) and NH<sub>4</sub>+ (Fig 3C), suggesting nitrification. This is because chemoautotrophs consume NO<sub>2</sub>- and NH<sub>4</sub>+ during nitrification to yield energy to fix DIC in the aphotic zone (REF), thereby reducing nutrient standing stocks in the water. We surmise that this 'depletion' may, theoretically, explain the observed negative correlation between nutrient levels and chemoautotrophic activity. In agreement, both ammonia and nitrite oxidizers were

found in the aphotic zone of all cruises (DNA level, discussion below), further highlighting their potential role as contributors to DCF in the southeast Mediterranean Sea. Nitrification measurements along with metagenomic tools, DCF (and BP) in aphotic water should be included in future dedicated studies to better refute or reinforce that oxidation of NO<sub>2</sub>- or NH<sub>4</sub>+ may provide chemoautotrophic prokaryotes the energy to fix DIC.

**Figure 3:** The relationship between aphotic DCF and BP (a), NO<sub>2</sub><sup>-</sup> (b) and NH<sub>4</sub><sup>+</sup> (c). Note that NO<sub>2</sub><sup>-</sup> and NH<sub>4</sub><sup>+</sup> was measured only during the 2023 cruises. The 95% confidence interval is shown in gray.

3.4 Potential chemoautotrophs based on microbial community structure - Analyses of 16S rRNA gene amplicons suggest that diverse bacteria and archaea may drive DCF in the aphotic southeast Mediterranean Sea (Fig 4A). Microbes found in our collected genetic material primarily include the order Nitrosopumilales ammonia-oxidizing archaea, which become dominant below DCM (up to ~30% read abundance near the bottom), corresponding to previous estimates based on *in-situ* fluorescent hybridization (De Corte et al., 2009). Nitrite-oxidizing Nitrospirales comprised ~1% read abundance at depths below 300 m. Among these lineages, analyses of indicator species identified seasonal variation in abundance of the orders Nitrosopumilales and Nitrososphaeria that were more prominent in the stratified period than in wintertime (p-value

Figure 4: (a) Read abundance of the 40 most abundant taxa (order level) from different depths offshore the Southeastern Mediterranean Sea. Surface samples are within the range of 1-5 m depth; near surface are between 20 and deep chlorophyll maximum (DCM); below DCM corresponds to 180-240 m depths; near bottom samples were taken circa 5 meters above the seafloor. Potential DCF microbes are shown in blue, Methylococcales methanotrophs are marked in magenta, and photosynthetic Synechoccocales are shown in green for reference. (b) A principal coordinates analysis

showing the differences in the structure of microbial populations based on 16S rRNA gene read mapping..

# **4 Conclusions**

Based on the conceptual model in Figure 5 that summarizes the annual microbial carbon exchanges in the southeast Mediterranean sea's offshore area, DOC supplied by LPP is negligible and cannot explain the 'high' BCD in the area, especially in the aphotic zone that is considered a 'microbial hotspot' with relatively high bacterial activity per cell (Hazan et al., 2018; Rahav et al., 2019). Our observations suggest that DCF may provide a substantial amount of the missing carbon, at least in the southeast Mediterranean Sea, while source/s for the remaining missing carbon are currently unknown and warrant more research.

Figure 5: A schematic illustration showing microbial carbon exchange in the southeast Mediterranean Sea. Values shown are the annual averages of LPP, DCF and BP. DOC exudation from LPP was assumed to be 20% of the rates, BR was calculated from BP and assuming BGE=0.20, Zooplankton's respiration (ZR) and excretion were compiled from Belkin et al. (2022). The numbers in brackets show the integrated values over 1750 m and expressed as g C m<sup>-2</sup> y<sup>-1</sup>. Regardless of the yet missing DOC sources, our results demonstrate the pivotal role DCF plays in compensating metabolic imbalances in carbon sources at the aphotic southeast Mediterranean Sea. Similarly, DCF was shown to be a significant process supporting microbial respiration and/or activity in aphotic layers (Baltar et al., 2009; Herndl et al., 2005; Yakimov et al., 2011), as well as in hydrothermal vents (Mattes et al., 2013) and cold seeps (Nakagawa et al., 2007). Note that, while our estimates of DCF contribution to aphotic BCD are based on widely accepted assumptions, they are subject to some uncertainties, particularly regarding BGE that may be changed on both spatial and temporal scales, as well as the fraction of DOC exported from the photic zone that may also change between seasons and water provinces (see discussion below). These uncertainties underscore the need for more precise and region-specific measurements of BGE and DOC fluxes to better constrain the role of DCF in deep ocean carbon cycling. Another potential uncertainty in measuring aphotic metabolic rates such as DCF or BP lies in the unclear effects of hydrostatic pressure on the activity of bulk microbial communities (Riebesell et al., 2009; Tamburini et al., 2013). Laboratory-based manipulations that do not account for in situ pressure conditions may alter DCF rates, potentially misrepresenting the actual contribution of chemoautotrophs to aphotic

BCD. This highlights the urgent need for more detailed investigations into how hydrostatic pressure influences microbial activity in the deep ocean.

Additionally, despite the contribution of DCF to the DOC pool (taking into account the uncertainties associated with it discussed above), as well as the other sources, very little of fixed carbon as particulate organic matter (POC) ends up in sediment traps located above the seabed (2-6%). This suggests that most of the fixed carbon arriving from DCF (as well as LPP other potential sources) is recycled in the water column and does not reach the seabed. The rapid microbial recycling of nutrients was mostly investigated in the photic layer of the southeast Mediterranean Sea (e.g., PO<sub>4</sub>, Thingstad et al., 2005) and little is known about the processes, which are often cryptic (e.g., NO<sub>2</sub>-oxygenation), occurring in the aphotic layers.

Understanding the 'dark end' of the biological pump in oligotrophic oceans, which plays an important (yet variable) role in oceanic carbon cycling and sequestration, will require a multidisciplinary approach that take into account all the uncertainties discussed above in light of our (and others) observations of DCF, especially in the context of ongoing and significant changes in the marine environment. Our study supports the need for adding DCF measurements to global carbon budgets as also mentioned by Baltar and Herndl (2010).

## Data availability statement

The raw physiological data used to generate figures 1-3 can be downloaded from the PANGEA repository: Reich, Tom (2025): Primal, chemo and bacterial productivity coupled with inorganic nutrient concentrations from an off shore, outgoing transect cruises: <a href="https://doi.org/10.1594/PANGAEA.975231">https://doi.org/10.1594/PANGAEA.975231</a>. Hydrological data can be

| 427                                                                   | downloaded from <a href="https://isramar.ocean.org.il/isramar2009/">https://isramar.ocean.org.il/isramar2009/</a> . Amplicon sequencing data                                                                                                                                 |
|-----------------------------------------------------------------------|------------------------------------------------------------------------------------------------------------------------------------------------------------------------------------------------------------------------------------------------------------------------------|
| 428                                                                   | are available as NCBI SRA archive project number PRJNA1215023.                                                                                                                                                                                                               |
| 429                                                                   |                                                                                                                                                                                                                                                                              |
| 430                                                                   | Author contribution                                                                                                                                                                                                                                                          |
| 431                                                                   | TR: Formal experimental work and data analyses; Investigation; Data Curation;                                                                                                                                                                                                |
| 432                                                                   | Visualization; Writing – Original Draft Preparation. NB: Resources; Writing –                                                                                                                                                                                                |
| 433                                                                   | Review & Editing. GSV: Investigation; Writing – Review & Editing. HH:                                                                                                                                                                                                        |
| 434                                                                   | Investigation. MRB: Investigation; Writing – Review. IBF: Conceptualization;                                                                                                                                                                                                 |
| 435                                                                   | Funding Acquisition; Supervision; data interpretation, Writing – Review & Editing.                                                                                                                                                                                           |
| 436                                                                   | ER: Conceptualization; Formal Analysis; Funding Acquisition; Supervision;                                                                                                                                                                                                    |
| 437                                                                   | Visualization; Writing – Review & Editing.                                                                                                                                                                                                                                   |
| 438                                                                   |                                                                                                                                                                                                                                                                              |
| 439                                                                   | Competing interests                                                                                                                                                                                                                                                          |
|                                                                       |                                                                                                                                                                                                                                                                              |
| 440                                                                   | The authors declare that they have no conflict of interest.                                                                                                                                                                                                                  |
| 440<br>441                                                            | The authors declare that they have no conflict of interest.                                                                                                                                                                                                                  |
|                                                                       | The authors declare that they have no conflict of interest.  Acknowledgements                                                                                                                                                                                                |
| 441                                                                   |                                                                                                                                                                                                                                                                              |
| 441<br>442                                                            | Acknowledgements                                                                                                                                                                                                                                                             |
| <ul><li>441</li><li>442</li><li>443</li></ul>                         | Acknowledgements  We would like to thank the University of Haifa for supporting T. Reich, and the                                                                                                                                                                            |
| <ul><li>441</li><li>442</li><li>443</li><li>444</li></ul>             | Acknowledgements  We would like to thank the University of Haifa for supporting T. Reich, and the Helmholtz funded International Laboratory; The Eastern Mediterranean Sea Centre-                                                                                           |
| <ul><li>441</li><li>442</li><li>443</li><li>444</li><li>445</li></ul> | Acknowledgements  We would like to thank the University of Haifa for supporting T. Reich, and the Helmholtz funded International Laboratory; The Eastern Mediterranean Sea Centre-An Early-Warning Model-System for our Future Oceans: EMS Future Ocean                      |
| 441<br>442<br>443<br>444<br>445<br>446                                | Acknowledgements  We would like to thank the University of Haifa for supporting T. Reich, and the Helmholtz funded International Laboratory; The Eastern Mediterranean Sea Centre-An Early-Warning Model-System for our Future Oceans: EMS Future Ocean Research (EMS FORE). |

| 450 | Statements and Declarations                                                      |
|-----|----------------------------------------------------------------------------------|
| 451 | The authors declare that they have no conflict of interest.                      |
| 452 |                                                                                  |
| 453 | Funding                                                                          |
| 454 | This work was supported by the German-Israeli Foundation for Scientific Research |
| 455 | and Development 2016021 and The National Monitoring Program of Israel's          |
| 456 | Mediterranean Waters.                                                            |
| 457 |                                                                                  |

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
