# Peer review of "(untitled)"

_EGUsphere, 2025_

## Author Comment (AC3)

17.07.2025

To: Prof. Damian L. Arévalo-Martínez

Editor, Ocean Science

We are pleased to resubmit our revised manuscript entitled "*Dark inorganic carbon fixation contributes to bacterial organic carbon demand in the oligotrophic Southeastern Mediterranean Sea*" (egusphere-2025-1445). We would like to thank the editor and the reviewers for their time and constructive feedback. We have carefully addressed all comments raised, and revised the manuscript accordingly.

Below please find our detailed reply to each of the comments/critiques (in bold).

Respectfully,

Tom Reich (on behalf of all co-authors)

**Editor**

Again many thanks for your submission to Ocean Science. Two experts have reviewed your manuscript and although overall they see your work as a potentially important contribution, both have raised several issues. I kindly ask you to address all these issues in your authors' replies. In doing so, I encourage you to disregard the sentence from Reviewer # 1 that includes the word "sloppy" as this is not constructive. Likewise, be reminded that you are not obliged to include all references indicated by the reviewers, unless there are valid reasons for it. Soon you will receive instructions regarding the next steps of the review process.

**Response: We thank the editor for the positive view of our manuscript. We have addressed the relevant/constructive comments and suggestions below, while leaving unconstructive wording etc. Most of the comments made by both reviewers significantly improved the readability and clarity of the revision, and we hope you will now find the manuscript suitable for publication.**

**Reviewer 1**

Overall, the manuscript has some value because of the value of the data provided (not very common to see this number of dark dic fixation measurements in relation to primary production rates). However, the writing is a bit sloppy, and the novelty of the study is not as high as suggested by the authors. This is because, currently, this manuscript is critically missing some very relevant citations, which need to be included, so that the readers get a more accurate understanding of the novelty of this manuscript (see specific comments below). Also, several of the main claims are not supported by data of this manuscript (see also specific comments below).

**Response: We thank the reviewer for the comments and suggestions made, to which we have addressed in full below. Specifically, we added more information on the hydrological variables, added integrated averages of DCF and other metabolic processes, included additional details on the calculations made to account for the role of DCF in supporting aphotic BCD, 'toned down' some of the statements made and comprehensively revised the conclusion section, adding more discussion to better address existing uncertainties, knowledge gaps, and future research directions. As suggested, we now also consider more studies to back up our claims and justify our observations (e.g., Riebesell et al., 2009; Tamburini et al., 2013; Agogué et al., 2008; Santinelli et al., 2013; Civitarese et al., 2010; 2023; Moutin and Raimbault, 2002; Herndl and Reinthaler, 2013; Zweifel et al., 1993; Mena et al., 2019). To further support our data, in answer to the reviewer's critique we provide a detailed explanation below to clarify and justify our interpretations.**

l.36-48- one key citation, that exactly looked on the contribution of DIC fixation to bacterial carbon demand is missing is this work, and should be included throughout this manuscript (Baltar, …Herndl, et al., 2010 GRL, doi:10.1029/2010GL043105: "Significance of non-sinking particulate organic carbon and dark $CO_2$ fixation to heterotrophic carbon demand in the mesopelagic northeast Atlantic". This shows that the idea of looking into the contribution of dic fixation to MCD is not novel from the current manuscript, but has been explored before.

**Response: This manuscript contributes valuable insights to the expanding body of knowledge on DCF in oligotrophic ocean regions, particularly from the rapidly warming Eastern Mediterranean Sea. We acknowledge the seminal work of Federico Baltar and Gerhard**

**Herndl and have now included the citation of this recommended paper (L. 37) along with others which we find useful/relevant.**

- Related to this point, another key citation that is surprisingly not included in this manuscript, is Baltar & Herndl, Biogeosciences 2019 (https://doi.org/10.5194/bg-16-3793-2019): "Ideas and perspectives: Is dark carbon fixation relevant for oceanic primary production estimates?". This paper did actually raise awareness about the relevance of DIC fixation and quantified how it compares to primary production estimates in the ocean. Thus, to be fair with previous research, and fair and accurate to the readers, the findings of the current manuscript should take into consideration what others have found before. For example, when looking into the main findings of this manuscript, mentioned in the last paragprah of the Introduction: this point in l.72-73 ("Our results demonstrate that DCF rates cannot be negleted (contrary to past conventions, Nielsen 1952))" was already concluded/mentioned in the Baltar & Herndl 2019 paper. In l. 74- 75. The other outcome ("We also show that DCF substantially contributes to bacterial carbon demand (BCD)") was also already shown by Baltar etc a 2010 GRL (although in the current paper is focused on the Med Sea, and in the case of Baltar et al 2010 it was focused on the Atlantic; but still, this information cannot be neglected).

**Response: We fully agree with the reviewer's comment and recognize this important work that should be cited. Baltar and Herndl (2019) is a pivotal study that has helped shape the framework of our current research. We now refer to this paper in our revised manuscript (L. 48).**

L. 99. specify what type/material of bottles. This is relevant for light productivity.
- what was the temperature of the incubation? relative to in situ conditions. This is very important due to the strong shift in temperature from 0 to >1500 meters.

**Response: Details on bottle material and incubation setup were added to the Methods section (please see below citation). While ambient light levels were maintained during the incubations, the water temperatures reflected surface conditions only (~19-20 °C in winter and ~28-29 °C in the summer cruises). We acknowledge that temperature differences between surface and deep-water incubations may alter the LPP or DCF rates measurements. This is particularly relevant during the summer cruises and to a lesser extent in winter where**

the temperature differences are relatively small. Note, however, that this incubation setting was the best available practice at the time of the cruises, like in many oceanographic studies. Preliminary comparisons between our incubation setup and in situ incubations (where bottles were deployed at their respective depth on a mooring line) showed negligible differences, falling within the expected range of measurement variability (see below). Similarly, Letelier et al. (1996) found that extended on-deck incubations, like those used in our study, may moderately underestimate primary productivity (maximal ~20%). Importantly, even if some degree of underestimation (or overestimation) occurred, it is unlikely to alter the observed vertical patterns or the calculated contribution of DCF to total primary production. This methodological issue was broadly discussed in a recent study from the Eastern Mediterranean Sea by Reich et al., (2022 DSRII). Please see also the figure and summary table below showing the comparison between the '*on-deck*' vs. '*in-situ*' primary productivity rate measurements.

[Figure]

| Station | PP (mg C m$^{-2}$ d$^{-1}$) | | |
|---|---|---|---|
| | *In-situ* | *On-deck* | Difference (%) |
| 1 | 53.45 | 44.63 | -18 |
| 2 | 50.04 | 48.45 | -3 |

The following clarifications were added:

"The bottles were maintained in on-deck incubators covered with a gradient of neutral mesh simulating the irradiance intensity (no change in spectrum) at 100%, 50%, 10%, 1%, and 0.1% of surface light intensities or under complete dark conditions (Belkin et al., 2022; Reich

et al., 2024). Incubators were kept at constant ambient surface temperatures (~19-20 °C in winter and ~28-29 °C in the summer cruises). We acknowledge that temperature differences between surface and deeper depths may alter the LPP or DCF rates measurements, especially during the summer when the water column is stratified. While *in situ* measurements may offer more precise rate estimates, they are generally impractical during research cruises that involve sampling at multiple locations and times throughout the day and night. Nevertheless, preliminary comparisons between the incubation setup used here versus *in situ* incubations using a mooring line showed negligible differences in primary productivity, falling within the expected range of measurement variability (see also Reich et al., 2022)…" (L. 105-117).

L.119. What 100 nM was used? seems quite high for an oligotrophic site.

- the number of replicates and blanks is currently missing; needs to be included.

**Response: We added 10 nmol/L of hot leucine and not 100 nmol/L, as used in previous studies from the area (e.g., Van Wambeke et al., 2002; Tanaka et al., 2007; 2011; Belkin et al. ,2022). We thank the reviewer for finding this typo.**

**We also added more information on the number of replicates used: "Triplicate samples per depth (1.7 ml) were incubated in the dark with 10 nmol/L …" (L. 136-137).**

- The methods are missing a detailed description of how the BCD were calculated (and the contribution of DCF to BCD). This is particularly relevant because respiration was not directly estimated in this study, but it was indirectly derived.

**Response: Bacterial respiration was calculated assuming an average open-ocean BGE of 20% (Herndl and Reinthaler, 2013) following the following equation:**

$$BGE = \frac{BP}{BP + BR}$$

**This estimate of the BGE is in the range previously reported from the Mediterranean Sea spanning from 0.21-0.29 (Zweifel et al., 1993). Once we estimated BR, we then calculated the BCD as the sum of heterotrophic microbial biomass production (BP) and bacterial respiration.**

**Note that we now added text discussing the uncertainties in this calculation (e.g., L. 168-175)**

- To properly distinguished the "mixed" and "staritifed" periods, the authors could also include the Temperature and Salinity (and/or density) plots, to show the level of stratification, and support their period clasification.

**Response: The mixed layer depth (MLD) was calculated using a temperature difference of Δ0.3 °C (Mena et al., 2019). This is a common practice in marine locations where thermal stratification is a dominant factor such as in the Eastern Mediterranean Sea (Ozer et al., 2017). The information was added: "...Three cruises were held during the stratified period and three during the winter mixing. To this end, the mixed layer depth was calculated using a temperature difference of Δ0.3 °C (Mena et al., 2019)..." (L. 83-85).**

**We believe that including additional temperature and/or density plots may introduce unnecessary figures that could divert attention from the main focus of the study, namely characterizing DCF and its contribution to PP throughout the water column. That said, all raw hydrographic data from the cruises are available in a freely accessible online repository. "...The raw hydrological data can be freely downloaded from https://isramar.ocean.org.il/isramar2009/..." (L. 93-94).**

Table 1. Not clear if those numbers are an interval of confidence or a range. This should be specified in the caption. And also, a measure of variability (SD, or SE,..) should be included.

**Response: The table shows integrated values from the photic and aphotic layers for each production parameter or contribution The values represent the minimum and maximum ranges observed across cruises, with the averages and its corresponding standard deviations (SD) provided in parentheses. Details added to the caption (L. 206-210).**

Also, the limitations of the assumptions in this ms (ie, "Assuming bacterial gross efficiency of 0.2 (Gasol et al., 1998) and that the available DOC for bacteria is 20% of the total primary productivity at the photic layer (Teira et al., 2003)" should be discussed in the discussion of this manuscript.

**Response: We thank the reviewer for this suggestion and added a discussion about the uncertainties of the calculations we used to estimate the role of DCF in supporting aphotic BCD. This include using a higher BGE value of 0.29 instead of 0.2 as reported from the Mediterranean Sea (Zweifel et al., 1993 MEPS) and use of a different factor to assess the contribution of exported DOC from PP to the aphotic zone (4% instead of 20%, Moutin and Raimbault, 2002). It is important to note that other studies examining the role of DCF in BCD have used similar calculations with the same assumptions and factors as ours, Chen et al,. (2020) for example. Therefore, we believe it is valuable to retain the original calculations to allow for comparisons with findings from other regions.**

**The following text was added: "...We note that these calculations are based on global averages and assumptions and therefore may be subject to some uncertainties. For example, BGE can vary between seasons and sites (del Giorgio and Cole, 1998). In the Mediterranean Sea, long-term measurements of BGE ranged from 0.21 (similar to our calculations and the global average used by Herndl and Reinthaler, 2013) to 0.29 (Zweifel et al., 1003). Under this circumstance, the contribution of DCF to the aphotic BCD increases to ~45% of the observations rather than ~35% when using BGE of 20%. Similarly, if we apply an exported DOC estimate of ~4% from the photic zone, as reported for the Ionian Sea/western Mediterranean (Moutin and Raimbault, 2002), the relative contribution of DCF to aphotic BCD would be even higher than in our current calculations, which assume ~20% DOC export (Teira et al., 2003). These uncertainties warrant further investigation in future studies, yet even when using conservative estimates for BGE and DOC export as done here, the contribution of DCF to aphotic BCD remains substantial..." (L. 256-270).**

- l. 207. Before it was mentioned they used the BGW by Gasol 1998, and here that they use the one from Doval 2001….this needs to be consistent.
 **Response: Reference changed for consistency (L. 211).**

-l. 212-215. All this is based on the contribution from DOC; but what about the contribution of POC?

**Response: Unfortunately, we do not have POC measurements. Nevertheless, we mentioned based on adjacent sediment trap data that despite the contribution of DCF to the DOC pool little of fixed carbon as POC ends up in sediment traps located at 1300 m (only 2-6%), suggesting that most of the fixed carbon arriving from DCF (as well as LPP other potential sources) is recycled in the water column and does not reach the seabed (L. 404-412).**

-l. 230-233. Here is mentioned that the nutrients "in the deep aphotic water were similar overall between periods (not shown)"; but in the methods section it was mentioned that nutrients were measured only in 2 of the 6 stations. Thus, it seems to me that this data is 'not show' because is not available, and therefore this statement cannot be sustained.

**Response: We revised the text for better clarity: "...Inorganic nutrients such as $PO_4^{3+}$ or $NO_3^-$ +$NO_2^-$ are unlikely to explain this variability as their ambient levels were similar between periods (https://isramar.ocean.org.il/isramar2009/). Alternatively, we surmise that..." (L. 280-287).**

-l 239-241. "therefore we surmise they have a preferential particle-attached". There is no data in this study to support this claim

**Response: The paragraph presents an hypothesis ("we surmise that...") stating that alterations in organic nutrients quality attributed to circulation pattern of deep water in the EMS may explain, at least partly, the interannual differences in DCF between March 2021-March 2022 and August 2022-August 2023. We say that several studies showed that organic nutrients promote chemoautotrophic activity. We revised the paragraph for better clarity: "...we surmise that differences in the bioavailability and concentration of sinking organic particles, possibly attributed by the BiOS (Bimodal Oscillating System) oscillation circulation of deep water between the Adriatic and Ionian seas, could potentially explain the higher aphotic chemoautotrophic activity in March 2021-March 2022 versus August 2022-August 2023. Where, this mechanism is known to influence the bioavailability of organic nutrients in the deep Mediterranean Sea by modulating deep-water circulation and ventilation patterns (Civitarese et al., 2010). These shifts affect the transport and residence time of organic matter (Civitarese et al., 2023), thereby potentially altering availability of organic nutrients to aphotic microbial populations, including to chemoautotrophs. In**

**support, recent studies from the northern Red Sea and South China Sea showed that DCF is limited by labile organic nutrients such as phosphonates and even carbon-rich molecules (Reich et al., 2024; Zhou et al., 2017)...” (L. 283-294).**

l.246. The microbial carbon pump citation is Jiao et al 2010, but not Herndl and Reinthaler 2013.

**Response: Sentence revised for clarity: “...Understanding how chemoautotrophs transform labile dissolved organic matter into refractory dissolved organic matter, which is an essential process in the ‘microbial carbon pump’, is crucial as it influences the efficiency of the biological pump (Jiao et al., 2010)...” (L. 303-306).**

l. 250-257. The authors mention that the negative correlation they find between ammonia and nitrite to DCF is indicative of an important metabolic pathway yielding energy for fix DIC in the aphotic zone. However, this does not make too much sense to me. The relation, for example, between ammonium oxidisers and DIC fixation should be positive and not negative, to support the relation of ammonium as an energy source of DIC fixation (see Agogue, ..Herndl et al., 2008, Nature). In fact, the same authors seem to agree with what I mention when they write right afterwards: “In agreement, both ammonia and nitrite oxidizers were found in the aphotic zone of all cruises (discussion below), further highlighting their potential role as contributors to DCF in southeast Mediterranean Sea”… meaning a positive relation (presence of both things: nutrients and DCF fix rates) would indicate that one relies on the other…In fact, this negative relation is probably more an indirect effect of the normal changes observed with depth in nutrient concentration in the water column. Thus, the argument in this paragraph is not supported by the data.

**Response: We disagree with the reviewer on this point. The presence of nutrients at the time of sampling represents their “standing stock” in the water column. Where, low concentrations imply they were consumed by surrounding microbes such as chemoautotrophs. Thus, one can ‘expect’ to see a negative correlation between ammonia or nitrite and DCF as a suggestive indication supporting nitrification. Similarly, the low $PO_4$ in the surface Mediterranean has been demonstrated to limit microbial populations both heterotrophic and photautotrophic (e.g., Krom et al., 2010, 2014; Thingstad et al. 2005) – an observation that was further examined in dedicated studies. From what we can see, Agogue**

et al., (2008) did not measure nutrients in their study, and thus the statement of the reviewer is unclear to us.

Moreover, the lack of any correlation between ammonia or nitrite and BP (e.g., Figure 3A) further suggests that their levels were not influenced by typical ("normal" as the reviewer wrote) biological processes and fluctuations by inhabiting microbes.

Lastly, it is important to note that the detection of ammonia and nitrite oxidizing taxa at the DNA level as we did (Figure 4) does not inform whether these organisms were active or inactive. Instead, it provides indirect and suggestive evidence for the potential of nitrification within the system. We encourage future studies to further explore the link between DCF and specific microbial groups such as archaea by targeting RNA-level expression and functional genes (e.g., *amoA*), as demonstrated by Agogué et al. (2008).

"...We stress that future studies should examine the link between DCF and specific microbial groups such as archaea by targeting RNA-level expression and functional genes (e.g., *amoA*), as demonstrated by Agogué et al. (2008)..." (L. 354-356).

We also revised the paragraph accordingly: "...Our results show a negative-linear relationship between DCF and $NO_2^-$ (Fig 3B) and $NH_4^+$ (Fig 3C), suggesting nitrification. This is because chemoautotrophs consume $NO_2^-$ and $NH_4^+$ during nitrification to yield energy to fix DIC in the aphotic zone (REF), thereby reducing nutrient standing stocks in the water. We surmise that this 'depletion' may, theoretically, explain the observed negative correlation between nutrient levels and chemoautotrophic activity. In agreement, both ammonia and nitrite oxidizers were found in the aphotic zone of all cruises (DNA level, discussion below), further highlighting their potential role as contributors to DCF in the southeast Mediterranean Sea. Nitrification measurements along with metagenomic tools, DCF (and BP) in aphotic water should be included in future dedicated studies to better refute or reinforce that oxidation of $NO_2^-$ or $NH_4^+$ may provide chemoautotrophic prokaryotes the energy to fix DIC..." (L. 311-323).

l. 265-276. The authors mention "Analyses of 16S rRNA gene amplicons suggest that diverse bacteria and archaea may drive DCF in the aphotic southeast Mediterranean Sea (Fig 4A)". However, the 16S analysis can only provide a general community composition and diversity of the

overall community, but it does not provide a direct link between these individuales and the DCF. In other words, it cannot indicate which of all the members of the community is performing DCF. This, their argument is also not supported by the data they provide.

**Response: We agree with the reviewers' note and added the following text: "...While this DNA-based community analysis provides insight into the potential contributors to DCF, it does not reflect a direct link as we cannot determine which of the identified microbes are responsible for the measured DCF rates. We stress that future studies should examine the link between DCF and specific microbial groups such as archaea by targeting RNA-level expression and functional genes (e.g., amoA), as demonstrated by Agogué et al. (2008)..." (L. 351-356).**

l. 281. I believe the citation Baltar et al 2022 might be outdated. This was a preprint but I think they published the final version of the paper in Nature Microbiology in 2023 or 2024. This should be updated.

**Response: The citation has be changed (L. 345)**

l.300-302. Based on all the assumptions and limitations of the present study I would not sue the expression "it is clear that" in this sentence.

**Response: The phrase has been removed as suggested.**

Overall the Conclusion paragraph seems quite weak. imprecise and unstructured. Also on of the final statements ("Our study highlights the need for adding DCF measurements to global carbon budgets"), was already mentioned by the Baltar,..,Herndl et al 2010, GRL, article. So, it should be also mentioned in here.

**Response: We have carefully revised the conclusion section to include a more detailed discussion of the uncertainties associated with the carbon balance presented in Figure 5. Additionally, we expanded the broader context by highlighting the importance of advancing DCF research in light of climate change and other oceanic shifts. We also rephrased the statement regarding the need to incorporate DCF into global carbon budgets and acknowledged relevant contributions from other studies (L. 368-419).**

**Reviewer 2**

The manuscript by Reich et al., "Dark inorganic carbon fixation contributes to bacterial organic carbon demand in the oligotrophic Southeastern Mediterranean Sea" presents novel measurements of light and dark primary productivity as well as bacterial productivity across the whole water column at 6 time points. The authors conclude that dark carbon fixation by chemoauthotrophy is a quantitatively relevant process that can account for a substantial portion, on some occasions even fully, of the bacterial carbon demand in aphotic layers of the water column. The authors address an important topic and provide a novel, comprehensive dataset on productivity rates. I therefore recommend the manuscript for publication, after the following comments have been addressed:

**Response: We thank the reviewer for his/her positive assessment of our manuscript. We revised and clarified the text as suggested and addressed all comments and edits thoroughly.**

Discussion: While the discussion touches on the most relevant points, it is in part missing depth and structure. I recommend restructuring it by using subheadings. Several questions remain open, e.g., what is the reason for the spatial variation of DCF across the water column (Fig. 3)? What could be limiting factors for each of the rates presented, that would explain their spatiotemporal pattern? A systematic discussion of these factors together with their uncertainties would substantiate the discussion.

**Response: We have elaborated the discussion and conclusion sections to better address the points raised by the reviewers. Specifically:**

**1) We added subheadings as suggested.**

**2) We added more explanations on the vertical and interannual variability of DCF.**

**"...The observed decrease in DCF rates with depth (Figure 2A) during the summer cruises may be partly attributed to a decline in the abundance of chemoautotrophs with depth. For example, Agogué et al. (2008) reported a decline in archaeal *amoA* gene copy numbers with depth in the eastern North Atlantic. Normalizing DCF rates to chemoautotrophic microbial cell abundance (or gene copy) could reveal a different vertical pattern. Another possible explanation for the decline in DCF rates with depth may be related to the weakening flux of sinking organic matter with depth that limits the substrates that fuel DCF (discussion below)..." (L. 189-196).**

**"...Inorganic nutrients such as PO₄³⁺ or NO₃⁻+NO₂⁻ are unlikely to explain this variability as their ambient levels were similar between periods (https://isramar.ocean.org.il/isramar2009/). Alternatively, we surmise that differences in the bioavailability and concentration of sinking organic particles, possibly attributed by the BiOS (Bimodal Oscillating System) oscillation circulation of deep water between the Adriatic and Ionian seas, could explain the higher aphotic chemoautotrophic activity in March 2021-March 2022 vs. August 2022-August 2023. Where, this mechanism is known to influence the bioavailability of organic nutrients in the deep Mediterranean Sea by modulating deep-water circulation and ventilation patterns (Civitarese et al., 2010). These shifts affect the transport and residence time of organic matter (Civitarese et al., 2023), thereby potentially altering availability of organic nutrients to aphotic microbial populations, including to chemoautotrophs..." (L. 280-292).**

3) We discuss the possible limiting factors for DCF:

**"...recent studies from the northern Red Sea and South China Sea showed that DCF is limited by labile organic nutrients such as phosphonates and even carbon-rich molecules (Reich et al., 2024; Zhou et al., 2017). Aphotic free-living chemoautotrophs are likely to encounter an increasingly refractory pool of dissolved organic matter for metabolism that may result in lower DCF rates, as shown in exported material through the water column (Santinelli et al., 2013). Particle-attached chemoautotrophs may have access to higher concentrations of organic substrates, and therefore we surmise they have a preferential particle-attached lifestyle in the deep ocean. The patchy nature of particulate matter sinking (Alkalay et al., 2024) and aggregate concentrations (Bar-Zeev et al., 2012) in the deep southeast Mediterranean Sea could also potentially explain the interannual variability in DCF between periods. Understanding how chemoautotrophs transform labile dissolved organic matter into refractory dissolved organic matter, which is an essential process in the 'microbial carbon pump', is crucial as it influences the efficiency of the biological pump (Jiao et al., 2010)..." (L. 292-306).**

4) We discuss in length about some uncertainties in measuring DCF and assessing its role to BCD. This includes:

**"...We note that these calculations are based on global averages and assumptions and are therefore may be subjected to some uncertainties. For example, BGE can vary between**

seasons and sites (del Giorgio and Cole, 1998). In the Mediterranean Sea, long-term measurements of BGE ranged from 0.21 (similar to our calculations and the global average used by Herndl and Reinthaler, 2013) to 0.29 (Zweifel et al., 1003). Under this circumstance, the contribution of DCF to the aphotic BCD increases to ~45% of the observations rather than ~35% when using BGE of 20%. Similarly, if we apply an exported DOC estimate of ~4% from the photic zone, as reported for the Ionian Sea/western Mediterranean (Moutin and Raimbault, 2002), the relative contribution of DCF to aphotic BCD would be even higher than in our current calculations, which assume ~20% DOC export (Teira et al., 2003). These uncertainties warrant further investigation in future studies, yet even when using conservative estimates for BGE and DOC export as done here, the contribution of DCF to aphotic BCD remains substantial..." (L. 256-270).

"...Note that, while our estimates of DCF contribution to aphotic BCD are based on widely accepted assumptions, they are subject to some uncertainties, particularly regarding BGE that may be changed on both spatial and temporal scales, as well as the fraction of DOC exported from the photic zone that may also change between seasons and water provinces (see discussion below). These uncertainties underscore the need for more precise and region-specific measurements of BGE and DOC fluxes to better constrain the role of DCF in deep ocean carbon cycling..." (L. 389-396).

"...Another potential uncertainty in measuring aphotic metabolic rates such as DCF or BP lies in the unclear effects of hydrostatic pressure on the activity of bulk microbial communities (Riebesell et al., 2009; Tamburini et al., 2013). Laboratory-based manipulations that do not account for in situ pressure conditions may alter DCF rates, potentially misrepresenting the actual contribution of chemoautotrophs to aphotic BCD. This highlights the urgent need for more detailed investigations into how hydrostatic pressure influences microbial activity in the deep ocean...." (L. 397-403).

"...We note that these calculations are based on global averages and assumptions and are therefore may be subjected to some uncertainties. For example, BGE can vary between seasons and sites (del Giorgio and Cole, 1998)..." (L. 256-259).

Conclusion: One of the conclusions is that the dark carbon fixation can not always cover the full bacterial carbon demand, requiring to assess additional carbon sources. I would recommend to add a description on uncertainties here, by systematically assessing the calculation of bacterial carbon demand and the parameters and assumptions (e.g. growth efficiencies, percentage of DOC exudation of primary productivity (see Thornton et al., 2014), …) that feed into it. Such a detailed uncertainty assessment would build more confidence in the final conclusion.

**Response: We thank the reviewer for this suggestion. We added more information on the uncertainties associated with our assumptions (e.g., see previous comment #4). Additionally, we recalculated the contribution of DCF to BCD using in situ BGE values (e.g., Zweifel et al., 1993) and a more conservative estimate of DOC export typical of the oligotrophic Mediterranean Sea. Notably, these revised calculations actually highlight an even greater significance of DCF to aphotic BCD.**

**"...In the Mediterranean Sea, long-term measurements of BGE ranged from 0.21 (similar to our calculations and the global average used by Herndl and Reinthaler, 2013) to 0.29 (Zweifel et al., 1993). Under this circumstance, the contribution of DCF to the aphotic BCD increases to ~45% of the observations rather than ~35% when using BGE of 20%. Similarly, if we apply an exported DOC estimate of ~4% from the photic zone, as reported for the Ionian Sea/western Mediterranean (Moutin and Raimbault, 2002), the relative contribution of DCF to aphotic BCD would be even higher than in our current calculations, which assume ~20% DOC export (Teira et al., 2003). These uncertainties warrant further investigation in future studies, yet even when using conservative estimates for BGE and DOC export as done here, the contribution of DCF to aphotic BCD remains substantial..." (L. 259-270).**

Carbon budget in Fig. 5: does the budget consider that LPP only occurs during daylight periods, whereas DCF does not follow such constraints? Where does the DOC from bacterial production originally comes from (I would assume there is rather bacterial uptake, or am I missing sth)?

**Response: LPP rates were calculated from samples incubated for 24 hours under ambient light conditions. Therefore, regardless of the differing constraints between DCF and LPP, the measured rates represent total daily production across a full light-dark cycle.**

**Regarding the second point, yes – you are correct. We believe the DOC is utilized by prokaryotes from the environment and is incorporated into biomass.**

**Both pathways are expressed in Figure 5 caption and associated text.**

Methods: Samples were incubated under "ambient light conditions" (l. 101) or under "ambient temperatures" (l. 121), but it is never specified what these conditions are. A table including all sample depth, the respective light and temperature condition, together with the final data is required for a comprehensive description. Please also comment on the fact that they were not incubated under in-situ pressure, especially in light of recent publications (Amano et al., 2022).

**Response: Details on sample incubation conditions (temperature, light) were added to the 'Material and methods' section to better describe the protocol used in this study. In this respect, the ambient light levels were maintained during the incubations using mesh screens that simulated 0-100% of surface irradiance (without altering the light spectrum), the seawater used for temperature regulation were collected from the surface layer. As a result, it did not reflect the cooler temperatures found at deeper depths, which could potentially influence metabolic rate measurements. Preliminary comparisons between on-deck incubations as done here and *in situ* incubations where bottles were deployed at their respective depths on a mooring line show that primary productivity only slightly altered between approaches. The changes were well within the expected variability (<20%). Importantly, even if some degree of underestimation or overestimation occurred, it is unlikely to affect the vertical trends or the calculated contribution of DCF to total primary production. This methodological consideration has also been extensively discussed in our recent study from the Eastern Mediterranean by Reich et al. (2022, DSR II). For reference, please see the figure and summary table above in our reply to Reviewer 1.**

**We discuss this potential cavate in length in the revised manuscript:**

"...Incubators were kept at constant ambient surface temperatures (~19-20 °C in winter and ~28-29 °C in the summer cruises). We acknowledge that temperature differences between surface and deeper depths may alter the LPP or DCF rates measurements, especially during the summer when the water column is stratified. While *in situ* measurements may offer more precise rate estimates, they are generally impractical during research cruises that involve sampling at multiple locations and times throughout the day and night. Nevertheless, preliminary comparisons between the incubation setup used here vs. in situ incubations using a mooring line showed negligible differences in primary productivity, falling within the expected range of measurement variability (see also Reich et al., 2022)...." (L. 108-117).

We acknowledge that incubating deep-water microbial communities under surface pressure conditions may affect rate measurements. However, this approach represented the most practical and widely used method available during the cruise, consistent with many other oceanographic studies (many of them in high impact journals from recent years). We believe this limitation warrants separate, dedicated investigations using specialized pressure-chambers and other dedicated equipment. This caveat is addressed and discussed in the revision:

"…Another potential uncertainty in measuring aphotic metabolic rates such as DCF or BP lies in the unclear effects of hydrostatic pressure on the activity of bulk microbial communities (Riebesell et al., 2009; Tamburini et al., 2013). Laboratory-based manipulations that do not account for in situ pressure conditions may alter DCF rates, potentially misrepresenting the actual contribution of chemoautotrophs to aphotic BCD. This highlights the urgent need for more detailed investigations into how hydrostatic pressure influences microbial activity in the deep ocean…" (L. 397-403).

Lastly, a link to the full hydrographical data (including CTD, fluorometer, dissolved O2 etc.) is now provided.

Lateral transport: The profiles were discussed mainly in the context of in-situ data. However, they are taken in a fluid and dynamic environment. A short paragraph discussing lateral aspects as a driver of variability would help to set the results in context. For example, what is the main flow direction of the water at the location of the profile? How variable are physicochemical parameters in the horizontal direction (e.g. from satellite data?)?

**Response: We briefly address the potential influence of lateral transport of organic matter on aphotic waters, as indicated by data from nearby sediment traps (Alkalay et al., 2024), which may help explain some of the observed variability, including the elevated aphotic DCF rates between winter and summer. Additionally, we hypothesize that the oscillating deep-water circulation between the Adriatic and Ionian seas (known as the BiOS) could account for the interannual differences in DCF observed between March 2021–March 2022 and August 2022–August 2023. However, we believe that expanding the discussion to include broader oceanographic features, such as main current systems or coastal surface water intrusions (Efrati et al., 2012), would be too speculative given the scope and data limitations of our study.**

**"...The patchy nature of particulate matter sinking and lateral transport during wintertime (Alkalay et al., 2024) and aggregate concentrations (Bar-Zeev et al., 2012) in the deep southeast Mediterranean Sea could also potentially explain the interannual variability in DCF between periods..." (L. 300-303).**

**"...we surmise that differences in the bioavailability and concentration of sinking organic particles, possibly attributed by the BiOS (Bimodal Oscillating System) oscillation circulation of deep water between the Adriatic and Ionian seas, could explain the higher aphotic chemoautotrophic activity in March 2021-March 2022 vs. August 2022-August 2023..." (L. 283-290).**

Minor comments:
37: remove "pool"
**Response: The word has been removed as suggested.**

39: throughout the manuscript, there are some occasions of vague formulations, e.g. "exported through photosynthesis". Please doublecheck for vague formulations throughout the manuscript.

**Response: The phrase has been changed "fixed by photoautotrophs via photosynthesis and exported from the euphotic zone…" (L. 39).**

162, "than" instead of "then"

**Response: Corrected.**

168: Results of BCD are presented here, but it is not described in the methods how this was calculated. I recommend to briefly provide the equation in the methods section.

**Response: Information about BGE and BCD was added (L. 168-175).**

181-186: This sentence is hard to read, consider splitting it into several parts.

**Response: The paragraph was revised for a better clarity: "...The higher aphotic DCF in the mixed vs. the stratified periods may be related to more bioavailable carbon that is transported from the photic layer as marine snow and supplies organic carbon to heterotrophic activity in the winter (coinciding higher LPP). However, given the oligotrophic nature of the southeast Mediterranean Sea (Berman-Frank and Rahav, 2012; Reich et al., 2022), most of the organic carbon (both particulate and dissolved originating from LPP) is recycled within the photic layer. Only a small fraction fluxes down to the aphotic zone and has been recorded in sediment traps (Alkalay et al., 2024)..." (L. 215-222).**

207: "literature" instead of "literary"

**Response: Corrected.**

212-213: a word is missing in this sentence, making it difficult to understand

**Response: The sentence was revised for a better clarity: "Under these circumstances, exudation of DOC from primary productivity at the photic layer estimated as 20% of the rates (Teira et al., 2003) equals to ~1-130 mg C m-2 d-1..." (L. 248-249).**

330-333: the sentence about "operational opportunities" comes a bit out of the blue here and is not connected to the original scope of the study, I recommend deleting it.

**Response: As suggested, this part was removed.**

312: "DOC exudation from LPP was 20% of the rates.", for clarity, better to write "DOC exudation from LPP was assumed to be 20% of the rates.", since this was not measured.

**Response: As suggested, all of the above comments have been addressed.**

References

Thornton, D. C. (2014). Dissolved organic matter (DOM) release by phytoplankton in the contemporary and future ocean. European Journal of Phycology, 49(1), 20-46.

Amano, C., Zhao, Z., Sintes, E., Reinthaler, T., Stefanschitz, J., Kisadur, M., ... & Herndl, G. J. (2022). Limited carbon cycling due to high-pressure effects on the deep-sea microbiome. Nature Geoscience, 15(12), 1041-1047.

---

## Author Response (AR2)

To: Prof. Damian L. Arévalo-Martínez

Editor, Ocean Science

We would like to thank the reviewer and the editor again for the carful consideration of our manuscript (paper # egusphere-2025-1445) and the positive view of it (a 'Minor revision').

Below please find our detailed reply to the last few comments made by Reviewer 2 (in bold).

Respectfully,

Tom Reich (on behalf of all co-authors)

The authors answered the comments thoroughly. I do not have any major comments, but three minor ones remain:

1) L. 231 "excluding sporadic measurements" – can you give details which data points you excluded and why?

Response: We did not remove any BP data points from the dataset, nor did we exclude them from the statistical correlation analyses. The sentence was meant to emphasize that, apart from a few cases where aphotic BP values were relatively high, BP was generally low and of similar magnitude, showing no significant correlation with aphotic DCF.

The sentence was revised for better clarity: "...Aphotic BP rates were usually of similar magnitude and typically <0.1  $\mu$ g C L-1 d-1 (Fig 1C)..." (Lines 231-232).

2) Fig. 5: It still looks odd to me that bacteria add 35 g C m-2 yr-1 to the DOC pool and respire 144 g C m-2 yr-1, while there is no arrow indicating DOC uptake. If this figure is supposed to show microbial carbon exchange as indicated in the caption, I think it is missing a central process and should be adapted accordingly.

Response: We thank the reviewer for this comment. First, bacterial respiration was not directly measured in this study but was instead estimated from BP and BGE. We address

the associated uncertainties in detail and note that direct measurements are warranted in future studies (e.g., lines 398-404). In addition, the 'BP arrow' in the original illustration was incorrectly oriented. BP represents the utilization of DOC by prokaryotes rather than serving as a source of it. We thank the reviewer for highlighting this error, which has now been corrected in the revised figure.

Title: This should be doublechecked by a native speaker, but should't it read "Dark carbon fixation contributes to MEETING the bacterial organic carbon demand ..."? (also in line 24)

Response: The title was revised for a better clarity: "Contribution of dark inorganic carbon fixation to bacterial carbon demand in the oligotrophic Southeastern Mediterranean Sea" and wording was changed in the abstract "Here, we investigated the role of photic and aphotic dark inorganic carbon fixation rates (DCF) and its contribution to bacterial carbon demand in the southeastern Mediterranean Sea during the mixed and stratified periods." (Lines 22-25).